# Investigation of the Effect of Geometry Characteristics on Bending Stress of Asymmetric Helical Gears by Using Finite Elements Analysis

**Andromachi-Efsevia Zouridaki \* and George Vasileiou**

Section of Mechanical Design & Automatic Control, National Technical University of Athens, Athens 15773, Greece; gvasileiou@mail.ntua.gr

\* Correspondence: ezourida@central.ntua.gr; Tel.: +30-6944161784

**Abstract:** Asymmetric helical gears have been under investigation for more than two decades due to their inherent ability to handle greater bending loads than their spur counterparts of comparable size (i.e., the number of teeth and module). For this type of gear, only one side of each gear tooth in a geared mechanism is usually loaded (driving/driven side), whereas the other remains mostly unloaded (coast side). Due to the asymmetry of the tooth, a nonlinear model is used. For that reason, a numerical design procedure is introduced involving the geometrical and structural modelling of conjugate helical gear sets. This is accomplished with the tool of Finite Element Analysis (FEA) which is presented to the scientific literature. The basic geometry is initially generated in 2D and thereafter converted to a 3D shape using Boolean operations. The rigid body which is necessary for FEA software is produced from Computer Aided Design (CAD) software (SolidWorks). This paper is focused on the effect analysis of different geometry characteristics on bending loads. The effects on bending stress play a significant role in gear design wherein its magnitude is controlled by the nominal bending stress and the stress concentration due to the geometrical shape of the teeth. The analysis of this effect of the different geometrical characteristics in the load is presented in detail. Moreover, a comparison of the stresses that are developed between pairs with asymmetrical helical teeth by keeping one geometric parameter constant and modifying the values of the other two parameters is presented.

**Keywords:** asymmetric; helical gears; FEA; tooth profile geometry; bending stress analysis

---

## 1. Introduction

A prominent need in industrial transmission systems is designing compact mechanisms with high load handling capabilities. As a result, a lot of research has been directed towards increasing the load capacity of gears, therefore resulting in smaller transmission systems [1]. The type of gears used in such transmission systems are asymmetric gears. Asymmetric gears are suitable for applications with mechanisms that do not require reversing the direction of rotation. So far, they have been used in various commercial applications and their use will increase as load requirements increase in the industry. For extrapolating the concept of asymmetric spur gears to helical gears, an optimal design algorithm has been implemented for the production of a Computer Aided Engineering -ready (CAE) CAD model. A prerequisite for implementing a FEA method is the use of a 3D geometric model that will be compatible with a FEA software [2]. The theory used for the creation of the 3D geometric model is based on the generalization of the unified theory of gearing (Spitas et al. 2002) to asymmetric helical gears [3,4].

The unified theory of gearing is a three-dimensional computation theory where one of the following three elements is known: the profile of the generator wheel or the profile of the rule, or the contact

surface as well as the basic kinematic characteristics of the grade. By using this theory, the coordinates of the corresponding point of the profile of the contact gear can be calculated when the corresponding point of the one of the above three elements and the vertical vector N on the given surface at that point are known. The unified theory of gearing is three-dimensional, which means that it is not required to know the contact lines on the generator tooth profile, as is necessary in the Litvin theory for spur gear pairs.

The design of asymmetric gears is a new concept which requires a different set of parameters, such as two pressure angles, one for each profile of the tooth. The involute shape varies with the pressure angle, therefore affecting the contact zone of the gear pair and eventually resulting in changes in contact stress (Hertzian). The increase in pressure angle increases bending strength and decreases surface stress, however there is an upper limit of pressure angle value in order to avoid pointed teeth and a reduction in the contact ratio. In addition, the gear tooth fillet radius influences the upper limit of the pressure angle. Hence, for the design of asymmetric gears, there are some factors that must be taken into consideration. Some of these factors are critical such as tooth width, tooth form factor, stress concentration factor, load sharing factor and magnitude of moment acting on a single tooth [5,6]. In this paper, an effect analysis of different geometry characteristics to bending loads is presented. Moreover, a comparison of the results of the stresses that are developed between pairs with asymmetrical helical teeth by holding one geometric parameter stable and changing the other two parameters (e.g., the same module and different helix angles and different widths of teeth) is thoroughly analyzed.

## 2. Modelling of Asymmetric Helical Gears Using Matlab

This study is based on the generalization of the unified theory of gearing (Spitas et al. 2002) to asymmetric helical gears. For the algorithm, the set of input variables consist of the two involute angles, the helix angle, the module, and the cutter tip radius, tooth thickness, addendum and dedendum coefficients, which are commonly used in the design of gears industrially and coincide with the parameters used in symmetric gears. The output of the algorithm is a set of (x,y) points that belong to the analytic model of the gear without any approximations or numerical methods applied. These points are interpolated with splines, with respect to the continuity of the gear in the addendum cycle and the root cycle (Figure 1). The errors introduced by the interpolation are orders of magnitude smaller than those that stem from common manufacturing methods of gears. The 2D curve that is created from the set of points is extracted in order to produce the 3D model of the gear. By the 3D model, the standardized Boolean operation is applied, according to practices commonly applied to symmetric helical gears and the 3D CAE.

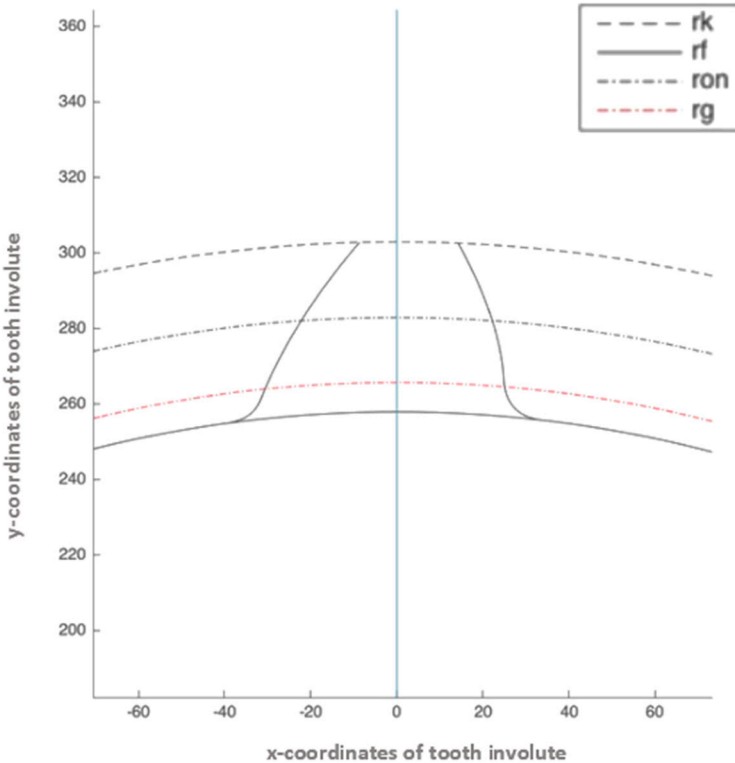

**Figure 1.** Asymmetric helical gear tooth profile generated from Matlab, where $r_k$ is the addendum circle radius, $r_f$ is the root fillet radius, $r_{on}$ is the pitch circle radius and $r_g$ is the base circle radius.

*The Basic Operations of the Algorithm*

i.      The algorithm receives, as an input, the module of the gear ($m_n$), the two pressure angles (driving and coast side), the helix angle, the number of teeth and any other basic gear tooth coefficients desired by the user;

ii.     It checks the gear for undercutting, backlash and interference;

iii.    It efficiently produces a set of points (x,y) that have been calculated analytically for the asymmetric gear, without resorting to numerical methods or applying some sort of approximation yet;

iv.    It produces either spur or helical asymmetric gears, according to the input of the user (when the helix angle = 0, a spur gear is produced);

v.      It interpolates the points with splines and produces the final model of the gear.

If all the appropriate conditions are observed, the algorithm produces the total view of the asymmetric helical gear tooth profile as shown in Figure 2.

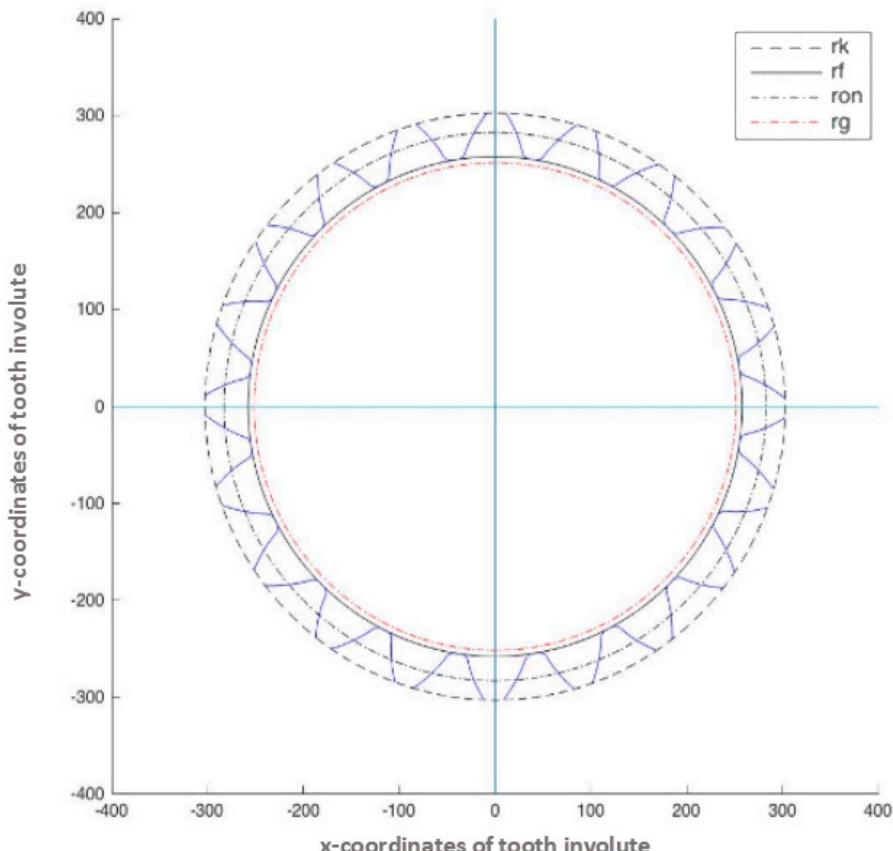

**Figure 2.** Total view of the involutes of asymmetric helical gear tooth profile.

## 3. FEA of Asymmetric Helical Gears

A 3D geometric model compatible with a commercial FEA software is produced by using a Matlab algorithm which was described above (Figure 3). By this algorithm, a series of points representing x- and y- coordinates of the tooth involutes with the origin located at the center of the gear are generated. The FEA method is used in order to extract conclusions for the behavior of the gear under stress. Bending stress, which is developed on asymmetric helical gear teeth, is an important parameter and it is a matter of great interest [7–11]. In this paper, the results of the stresses that are developed between pairs with asymmetric helical teeth are presented by holding one geometric parameter stable and changing the other two parameters (e.g., the same module and different helix angles and different width of teeth). To examine the bending stress in the gear pair, the maximum principal stress at the root on the tensile side of the tooth was used for evaluating the tooth bending strength of a gear and pinion.

The FEA software used was ANSYS in order to simulate the static and dynamic strength of asymmetric helical gear pair. The asymmetric helical gears which were designed and analyzed by using the ANSYS program were the pressure angles at the coast and drive sides, 14.5° and 20°, respectively. The asymmetric helical gear pairs were designed for the 5 and 10 mm modules. The helix angles that are compared in this paper are 20° and 30°. The face widths that are compared are 40, 50 and 60 mm. In the following figure, the parameters defined in the ANSYS program are presented.

With frictional adjustment, the two geometries in contact can bring shear stresses along with their interface before they begin to slide, one relative to the other.

This model defines an equivalent shear stress in which the sliding effect in geometry begins as a result of the contact pressure. Once the shear stress is overcome, the two geometries will slide, one relative to the other.

The frictionless setting is a one-way contact pattern, i.e., if separation occurs, the normal pressure equals zero. Thus, gaps in the model can be formed between the bodies depending on the load. This solution is

nonlinear because the contact area can change as the load is applied. The friction coefficient is considered to be zero, allowing the free slipping. The model should be clearly defined when this contact setting is made.

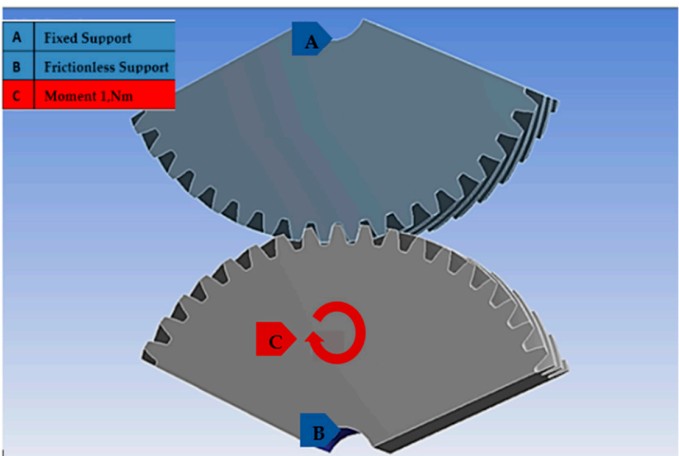

**Figure 3.** Parameters of ANSYS program.

ANSYS has the ability through this adjustment to balance the model statically. This ability ensures that there will be no variations in loads and in deformations in the case of an asymmetrical model. Every time the sliding is considered to be zero, a remeshing is done, because the problem is nonlinear.

Initially, a thick mesh of elements for the 3D model mentioned above is created in ANSYS. In order to optimize this mesh of elements, the parameters of the initial mesh are changed as detailed below. The edges are meshed with global element size, the edges are refined for curvature and 2D proximity and corresponding face and volume mesh is generated (the advanced Sizing Function was set to "off"). The mesh of asymmetric helical gear pairs consists of elements of 5mm size (Figure 4). The size transition was set to "Fast" (because of computational time). Inflation transition was set to "smooth" (which maintains smooth volumetric growth between each adjacent layer; total thickness depends on the variation of base surface mesh sizes). The chosen Inflation Algorithm (set to "pre") uses meshes which are set to create a five-layer inflation mesh. The method for patch conforming tetrahedrons is chosen. (Figure 5) The pinion was fixed while a load of 1Nm was applied on the gear.

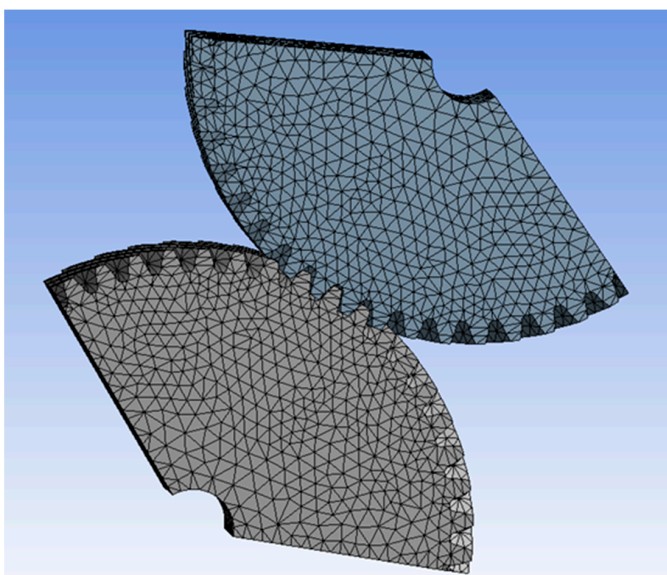

**Figure 4.** Asymmetric helical gear pair meshing.

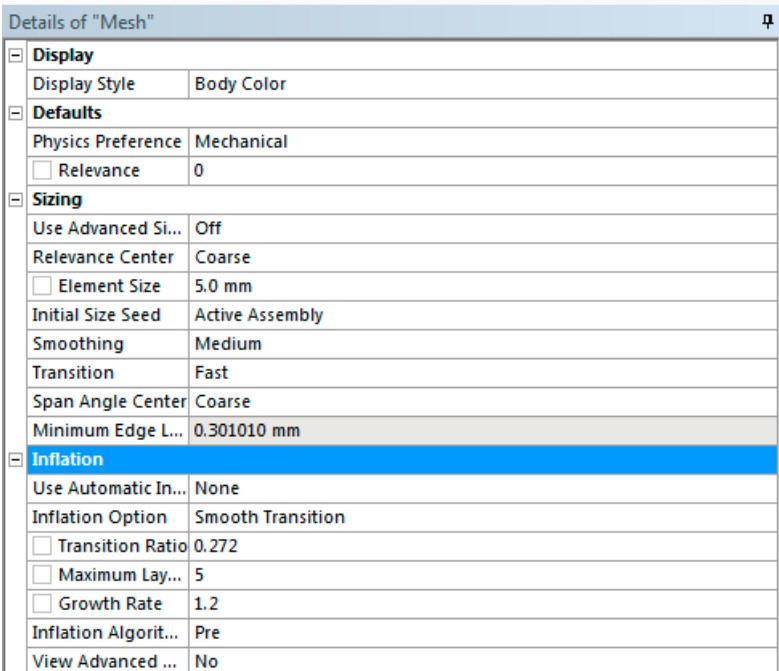

**Figure 5.** Details of "Mesh".

## 4. Comparative Study of the Stress on Asymmetric Helical Gear Pairs

In this section, a comparative study of the stresses which are developed between asymmetric helical gear pairs is presented by holding one parameter stable and modifying the other two parameters. Theoretical analysis has been performed on the asymmetric helical gear system using Matlab and a FEA analysis was performed by creating a model in SolidWorks and importing this file in ANSYS.

In this paper, the results related to the surface pressure resistance of asymmetric helical gear teeth through maximum von Mises stress and also the maximum principal stress at the root of asymmetric helical gear foot are presented. Both bending stresses and contact stresses in a helical gear pair depend on the helix angle, module and face width of the gear.

In this study a lot of different combinations took place. The following comparative diagrams (Figures 6 and 7) present the von Mises stress results. These results describe the resistance of asymmetric helical gears on surface pressure when the module is of a specific size (module of 5mm in Figure 6 and module of 10 mm in Figure 7), the helix angle is not changing (20°) and the face width is modified (40, 50 and 60 mm). From the results which follow when the face width increases, the resistance on surface pressure (stiffness) increases. Stiffness is calculated as the result of the division of the moment by the total deformation. In Figure 8, the results of another combination are presented. In this comparative diagram, the resistance of asymmetric helical gears on surface pressure is presented by keeping stable the module and face width and by modifying the helix angle. For greater helix angles, the stiffness of the asymmetric helical gears increases.

In Figures 9 and 10, the maximum principal stresses of asymmetric helical gears are presented. These diagrams represent the comparative diagrams of stress distribution at the root of the asymmetric helical gears (max principal stress). When the module is of a specific size and the helix angle is not changing, it is observed that, by increasing the face width, the stress distribution at the root of the asymmetric helical gears is decreased. There is also the corresponding stress distribution at the root of the asymmetric helical gears for bigger modules (Figure 9). The stress distribution at the root of the asymmetric helical gears decreases when the module is increased.

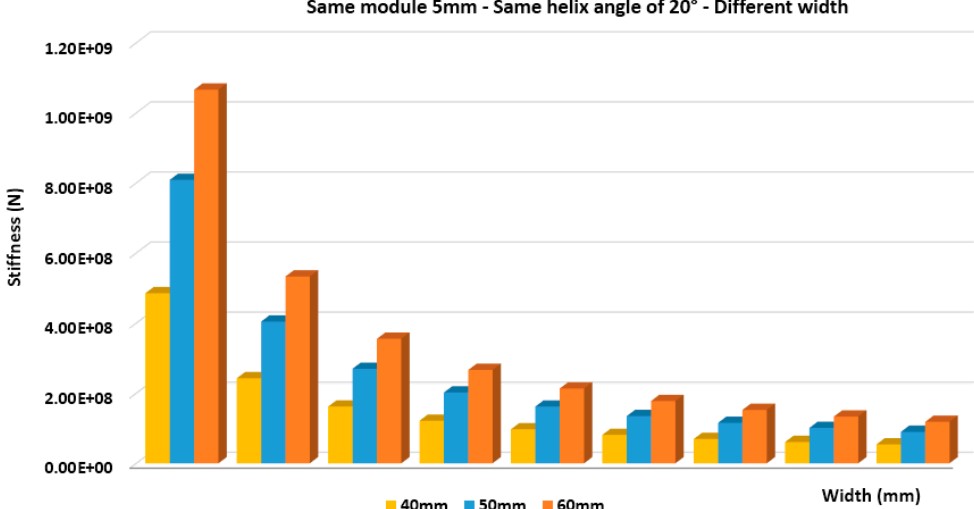

**Figure 6.** Gear stiffness for module 5mm and different face widths.

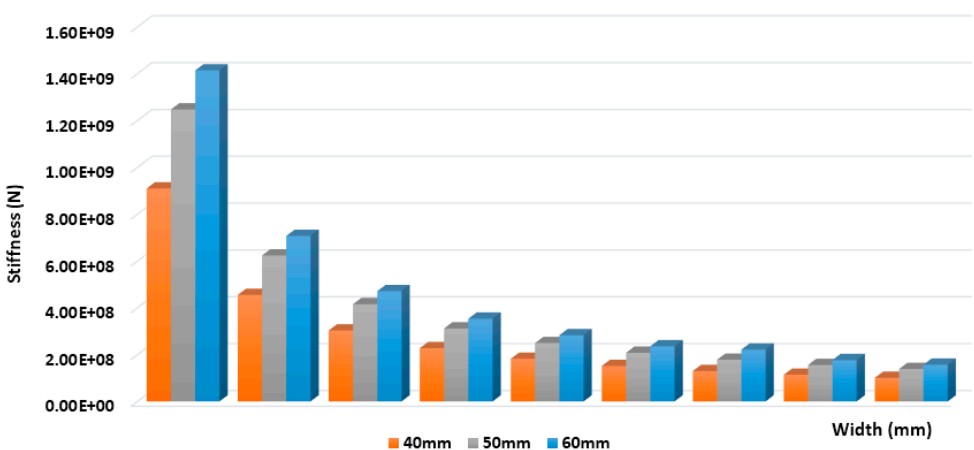

**Figure 7.** Gear stiffness for module 10 mm and different face widths.

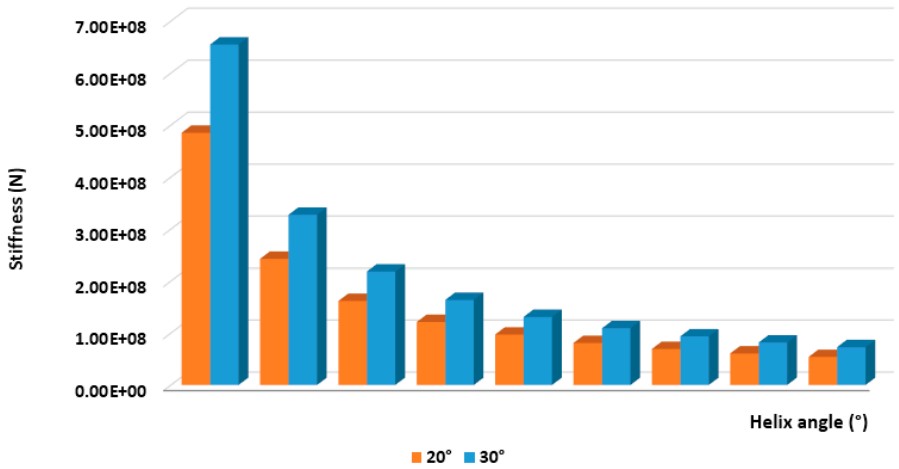

**Figure 8.** Gear stiffness for helix angle 20° and 30° for stable module 5mm and face width 40 mm.

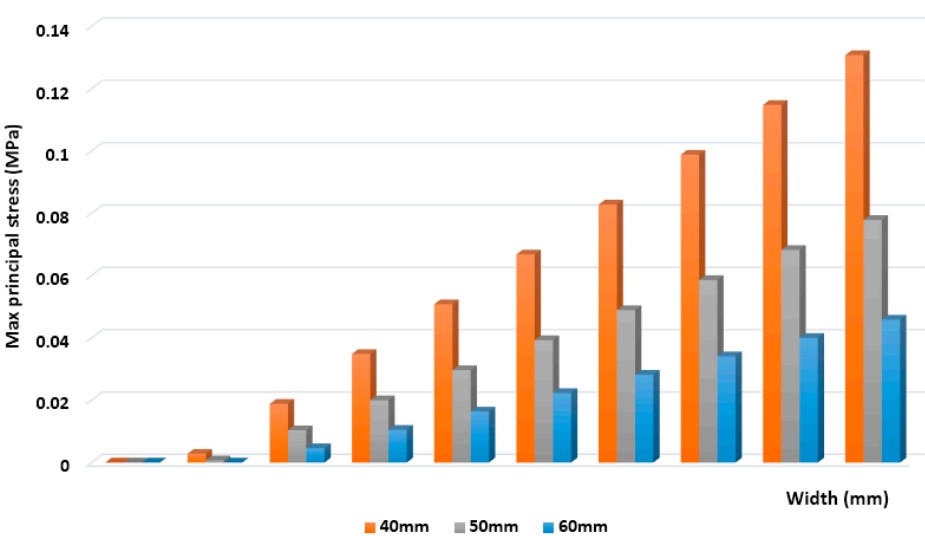

**Figure 9.** Maximum principal stresses for module 5m and different face widths.

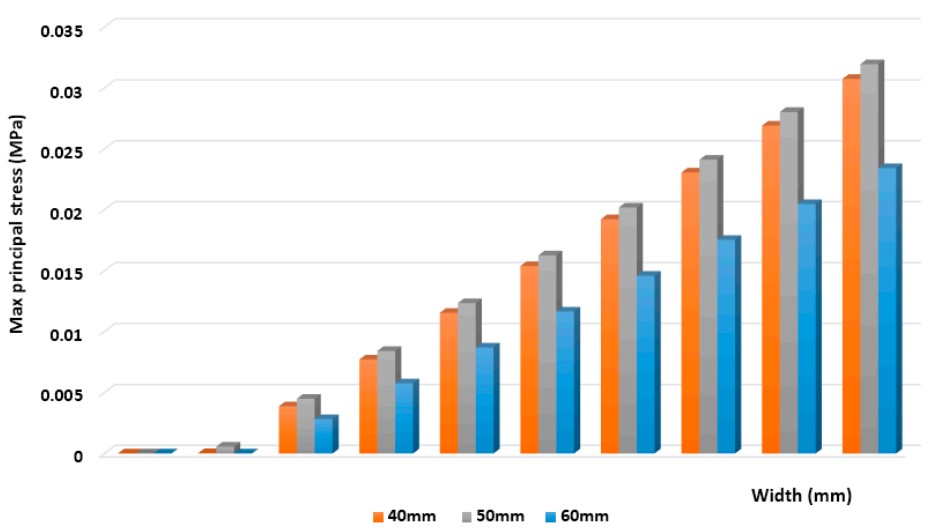

**Figure 10.** Maximum principal stresses for module 10m and different face widths.

## 5. Conclusions

In this paper, a geometric model of asymmetric helical gears was introduced and implemented. Theoretical analysis has been performed to an asymmetric helical gear system using Matlab and an FEA analysis was performed by initially creating a model in SolidWorks and importing it into ANSYS.

We also presented an effect analysis of different geometry characteristics on bending loads. Moreover, a comparison of the results of the stresses that are developed between pairs with asymmetrical helical teeth when holding one geometric parameter stable and modifying the other two parameters (e.g., the same module and different helix angles and different widths of teeth) is presented. As a result, it is excluded that von Mises stresses (i.e., the resistance of asymmetric helical gears on surface pressure) increase when the face width increases. By keeping the contact between the module and the face width and by modifying the helix angle, the resistance of asymmetric helical gears on surface pressure increases when the helix angle is increased. Furthermore, from FEA analysis, it was found that, by increasing

the face width, the stress distribution at the root of the asymmetric helical gears is decreased, while it decreases when the module is increased.

**Author Contributions:** Conceptualization, A.-E.Z.; methodology, A.-E.Z.; software A.-E.Z. and G.V.; validation, A.-E.Z.; formal analysis, A.-E.Z.; investigation, A.-E.Z.; resources, A.-E.Z. and G.V.; data curation, A.-E.Z. and G.V.; writing–original draft preparation, A.-E.Z.; writing–review and editing, A.-E.Z.; visualization, A.-E.Z.; supervision, A.-E.Z.; project administration, A.-E.Z.; funding acquisition. All authors have read and agreed to the published version of the manuscript.

**Funding:** This research received no external funding.

**Conflicts of Interest:** The authors declare no conflict of interest.

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
