# Peer review of "Investigation of the Effect of Geometry Characteristics on Bending Stress of Asymmetric Helical Gears by Using Finite Elements Analysis"

_computation, doi:10.3390/computation8010019_

Round 1

Reviewer 1 Report

This paper investigates the effects of geometrical characteristics on bending stress of asymmetric helical gears using finite element analysis procedure. However, the writing is poor and the paper lacks a detailed discussion of the simulation results. Some comments are given below:

  1. The writing should be improved as there are many unclear expressions and grammatic errors.
  2. The appearance of figures should follow an ascending order.
  3. In figure 1 and figure 2, the unit should be specified.
  4. In figure 3, the parameters used in ANSYS program are not shown.
  5. In figures 5, 6, 7, 8, and 9, what’s the meaning of horizontal axis?
  6. 6. The paper lacks a detailed discussion of the FEA results.

Reviewer 2 Report

The study presents the effect of the geometry of asymmetric helical gears.

The analysis was performed using the ANSYS FE code.

The following remarks should be taken into account:

  1. The title should be modified to something like :"An investigation of the effect of geometry characteristics on bending stress of asymmetric helical gears".
  2. The FE model should be more carefully described.
  3. How were the loads introduced ? Please discuss the issue by adding a dedicated drawing.
  4. Discuss the convergence of the FE model.
  5. Add the novelty of the present research over other existing studies.

Round 2

Reviewer 1 Report

It’s better to give the formula or definitions of stiffness (Figures 6, 7, 8), Max principal stress (in Figures 9, 10).

Author Response

I'd like to thank you for your comments which helped me to improve my paper.

Definition of stiffness calculation was added in lines 159-160

Max principal stress explanation was added in line 176.

Reviewer 2 Report

The authors implemented the suggestions of the reviewer, improving the content of the manuscript.

The present for of the article is now suitable for publication.

Author Response

I'd like to thank you for your comments which helped me to improve my paper.